# Planned, ongoing and completed tuberculosis treatment trials in Brazil, Russia, India, China and South Africa: a 2019 cross-sectional descriptive analysis

Lindi Mathebula [1,2] Lovemore Mapahla,[2,3] Dilyara Nurkhametova,[4] Liliya Eugenevna Ziganshina [4,5] Mikateko Mazinu,[3] Esme Jordan,[3,6] Duduzile Edith Ndwandwe [1] Tamara Kredo [1,7]

For numbered affiliations see end of article.

**Correspondence to**
Lindi Mathebula;
lindi.mathebula@mrc.ac.za

## ABSTRACT

Tuberculosis (TB) remains a deadly challenge globally and Brazil, Russia, India, China and South Africa (BRICS) are among the countries with the highest TB burden. The objective of this study is to identify and describe ongoing, planned and completed TB trials conducted in the BRICS countries registered in WHO-International Clinical Trial Registry Platform (WHO-ICTRP); to report selective outcome reporting by comparing primary outcomes in published trials with their prespecified outcomes in registry records and to evaluate the time to publication.

**Methods and analysis** We searched the WHO-ICTRP portal (20 January 2019) and the Russian Federation Registry (30 March 2019) to identify TB trials conducted in BRICS countries. We included only registered clinical trials conducted wholly in BRICS countries or with at least one recruitment centre in one of the BRICS countries that were investigating TB treatment.

**Results** The search of the WHO-ICTRP yielded 408 trials and additional 32 trials were identified from the Russian registry. Of those, 253 were included in the analysis. We found that 77 trials were multicountry trials, followed by trials in China (55), India (53), South Africa (34), Russia (23) and Brazil (11). 163 trials were registered prospectively, 69 retrospectively and 21 trials had no registration status. Most trials (207) evaluated TB treatment, followed by 29 behaviour change interventions, 13 nutritional supplementation, 4 surgical treatment and 2 assessing rehabilitation. Based on ICJME recommendation of publishing 12 months after completion of trial, we found that 156 trials were completed 12 or more months by date and 101 trials had publications. Thirty-one of the 101 trials with publication had evidence of selective outcome reporting. The median time to publication was 25 months (IQR 15–37) from the time of anticipated end date stated in the registry.

**Conclusion** TB trials conducted in BRICS countries are collaborative, mostly drug treatment oriented, potentially affecting policies. Selective outcome reporting remains a problem both for prospectively and retrospectively registered trials, only small fraction of which gets to publication.

## STRENGTHS AND LIMITATIONS OF THIS STUDY

⇒ This study addresses a research gap by describing and analysing planned, ongoing and completed tuberculosis (TB) treatment trials specific to middle-income countries with high TB burden: Brazil, Russia, India, China and South Africa.

⇒ We conducted an extensive search of relevant clinical trials registries including the WHO International Clinical Trial Registry Platform (WHO-ICTRP) and the Russian Trials Registry to ensure that no publicly available trial is missed.

⇒ The selection of trials and data extraction were conducted in duplicate to minimise bias and ensure rigour.

⇒ Since initial search on the 20 January 2019, there have been 20 more publications from included clinical trials. Studies published after our analysis are unlikely to change the description analysis we have performed.

⇒ As not all trials are registered or submitted to the WHO-ICTRP, some trials may have been missed.

## INTRODUCTION

Tuberculosis (TB) remains one of the most challenging and deadliest infectious diseases in the world ranking as the ninth cause of death worldwide.[1 2] Although TB mortality and incidence have been reported to be decreasing by 3% and 2%, respectively, most countries have been lagging behind in meeting the United Nations 2030 Sustainable Development Goal of ending the TB epidemic.[2] Brazil, Russia, India, China and South Africa (BRICS) countries have been reported to have a very high burden of TB, hosting 46% of all global TB cases and 40% of TB-related deaths.[3] Furthermore, antimicrobial resistance has led to increasing cases of Multi-Drug Resistant TB (MDR-TB) of which China, India and Russia account for more than half (56%) of the global burden.[3] In

addition to the Sustainable Development Goals launched in 2015 aiming to end TB by 2030, the WHO launched a strategy in 2014 called the WHO's End TB strategy, which is endorsed by the World Health Assembly, to eradicate the global TB epidemic by 2035.[3 4]

The WHO's End TB 2035 strategy aims to decrease TB-related deaths by at least 95% and an overall 90% reduction in TB incidence to 10 cases/100 000 or fewer with TB-affected households experiencing no costs of TB-related care.[5] One of the three pillars of the WHO End TB strategy is research and innovation. While research studies regularly estimate incidence, mortality rates and TB management approaches, new innovations of treatments are often not easily tracked.[2] To track TB research efforts, Lienhardt et al suggested a strategic plan that included mapping research funding involvement to determine where the research funding gaps are.[6] Another component of this plan is a Global TB Research Roadmap, which identified research gaps that should be closed in order to stop TB.[6] A study by Bai et al (2018) aimed to conduct a bibliometric analysis of research and development using publications to find out what contributions the BRICS countries make in the area for neglected tropical diseases including TB.[7] Another approach to identify research activity in different areas suggested by Viergever et al is to use data from registered trials in primary clinical trial registries.[8]

Mapping clinical trial activity through clinical trial registries provides an insight into where most research has been conducted and what type of research is being done or is planned and ongoing.[8–10] Clinical trials are primary research methods that are undertaken to study effects of new interventions such as treatment.[11] Randomised controlled trials (RCTs) have been described as the gold standard study design to inform policy and practice decisions regarding what interventions work better for improving healthcare.[12] However, studies showed that 37% of clinical trials presented in abstract and registered in trial registries never reach publication.[12 13] Furthermore, those that reach publication often omit negative results and deviate from prespecified outcomes leading to publication bias.[12–14] A systematic review looking at quality of outcome reporting in phase II studies of TB treatment showed evidence that these studies often had variations between reported outcomes and trial characteristics.[15] Research has suggested that for trials to be transparent, trialists should register their trials prospectively in a public platform indicating the aim of the study and the outcomes to be measured before the trial begins, to minimise selective outcome reporting (mismatch between registered and reported outcomes).[16]

Primary registries are online platforms that register and make public planned clinical trials and the data from different primary registries are aggregated and also made publicly available in the WHO-International Clinical Trial Registry Platform (WHO-ICTRP).[17] This then creates the necessary public accountability mechanism to ensure that planned trials and their results cannot be hidden regardless of their outcomes, which reduces the temptation to not publish results that are negative.[18] Prospective registration of trial in a primary registry means registration of a trial before it starts and is now widely recognised as a key strategy to increase research transparency. Prospective registration is supported by the Consolidated Standards of Reporting Trials (CONSORT) Statement and The International Committee of Medical Journals Editors (ICMJE), which includes many of the world's leading journals.[18–20] Retrospective registration of a trial means a trial that is registered after the first participant has been enrolled into the trial and is not a recommended practice as it diminishes research integrity according to the ICJME and CONSORT statement.[18–20]

BRICS countries have become major contributors to clinical research worldwide. Funding for research in the TB field and the number of publications has been seen to increase on a yearly basis having almost doubled in the past decade.[21] Research shows there is a likelihood of omitting or deviating from outcomes or measurement of outcomes when results are not favourable in trials including TB trials.[14 15] This may lead to overestimation of the efficacy of interventions and lead to adverse outcomes when implemented into policy. Proehl et al (2018) have indicated the need for transparency in clinical trials from trial registration to publication in order to effect a positive change in policy.[22] In light of this, the objective of this study is to identify and describe ongoing, planned and completed TB trials conducted in the BRICS countries registered in WHO-ICTRP and report any selective outcome reporting by identifying registered trials that have been published to compare reported outcomes with those prespecified in the trial registration. Mapping TB treatment trials will assist researchers to identify potential research gaps, sponsors and funders to identify funding gaps, patients to identify available treatments, policy makers to identify evidence-based interventions to end TB epidemic.

## OBJECTIVES

The main objective of the study is to identify and describe ongoing, planned and completed TB trials conducted in the BRICS countries registered in WHO-ICTRP and report any selective outcome reporting by identifying registered trials that have been published to compare reported outcomes with those prespecified in the trial registration.

### Hypothesis

The hypothesis of this study is that there is a difference between outcome-specified trial publications and those pre-specified in the trial registration.

## MATERIALS AND METHODS
### Study design

This is a cross-sectional study, in which we describe and analyse data from registered trials studying TB treatment

in BRICS countries found on WHOs clinical trial registry platform.

## Study sample

We aimed to include all registered trials planned for conduct in BRICS countries from inception of the WHO-ICTRP data repository in 2005 to the date of search. We searched the WHO-ICTRP portal on 20 January 2019 to identify TB trials conducted in BRICS countries using the following search terms: 'tuberculosis' AND 'Treatment' OR 'Therapy' OR 'Management' AND 'Country-name' that is, each of the BRICS countries. Two researchers (LindiM and LovemoreM) independently screened all records of the search and selected trials to be included. The Russian Federation does not have a primary registry that submits trials data to the WHO-ICTRP. Therefore, two researchers (DN and LEZ) from Cochrane Russia (based at Russian Medical Academy for Continuing Professional Education, the Russian Federation) searched the Russian trial registry, named the State Registry of Medicines (GRLS, grls.rosminzdrav.ru; 30 March 2019). This registry contains trial information on all clinical trials officially approved by the Federal Ministry of health. Since the Russian registry does not allow for an advanced search with Boolean operators, we searched for trials using the term 'tuberculosis' followed by a series of separate searches for each of the following antituberculosis medicines (search terms are available in online supplemental appendix 1).

We included only registered clinical trials regardless of status of the trial (ie, ongoing, planned or completed) conducted wholly in BRICS countries or with at least one recruitment centre in one of the BRICS countries that were investigating interventions for the treatment of TB. Trials were either RCTs, controlled clinical trials including cluster trials or multiple-arm trials. On retrieval of the search output, we removed studies that were observational from the sample and excluded duplicates and trials that were not eligible. WHO-ICTRP considers the first trial record registered to take precedence for trials registered in more than one primary register; however, we considered the trial record with the latest date and excluded the other records to ensure that we have a full record of the changes that were made from earlier trial registration.

## Patient and public involvement

We had no public or patient involvement in the conduct of this study.

After selection of the trials into the study, two researchers (LindiM, LovemoreM) independently used trial identification numbers to search for publications in PubMed and Cochrane Library. DN and LEZ searched the Russian electronic database of research publications eLIBRARY.RU using trial identification numbers and antituberculous medicines names. Publications were further identified in the trial record under the field 'URL

to publication' which is one of the 24 data items that is collected by WHO-ICTRP primary registries.

## Data extraction

### Trial records

We downloaded data from the WHO-ICTRP and imported into Microsoft Excel with the following data items: registry name, trial identification, initial application date, start date of trial, end date of trial, date of registration, registration status (retrospective/prospective), principal investigator, source of funding, type of treatment, primary and secondary outcomes, inclusion/exclusion criteria, age of participants and sample size among other baseline characteristics. Data from the Russian trial registry was manually entered in the Excel format under the same data items. Two researchers (LindiM and LovemoreM) independently reviewed data from the WHO-ICTRP manually and extracted further data points from individual records of included trials, similarly two researchers (LEZ and DN) from Russia independently and manually reviewed data from the Russian trial registry and extracted further data points from individual records. The researchers translated records that were in Russian language to English before sending data in Excel format.

### Publication extraction

Two researchers (LindiM and LovemoreM) independently extracted data from all published studies of the included trials in Excel for comparison. The following data items were extracted from the publication: title of the study, publication date, trial identification listed (yes/no), study design, age of participants, sample size and primary outcomes. We combined the data extracted by the two researchers from South Africa and two researchers from the Russian Federation for cleaning and coding. For selective outcome reporting we compared what was reported in the trial record as a primary outcome including the type of outcome and time points for the measurement of the outcome, to what was reported in the publication as a primary outcome. We used 'yes' if the authors reported the same outcome and 'no' if they reported a different primary outcome in the publication. We coded the data using variables in categories that unified the data (online supplemental appendix 1).

### Data analysis

Once entered, data were reviewed and cleaned to ensure that no data were missing, the completed and cleaned dataset was imported into the StataCorp 2015 for statistical analysis. All data were stored on the South African Medical Research Council network and backed up on a daily basis to ensure no loss of original data. Data were analysed using Stata statistical software. We used descriptive statistics and summarised data as counts (n) and frequencies (%) for categorical variables, and as medians with interquartile ranges for numerical variables.

For analysis of selective outcome reporting (ie, reporting outcomes in a publication that differ from those planned

in the trial protocol), we had two assumptions: (1) there may be a difference in selective outcome reporting when a trial is registered prospectively compared with retrospectively registered trials; and (2) there may be selective outcome reporting for trials that had multiple recruitment centres compared with those with single recruitment centres. To test our assumptions, we planned paired binary data test, which was relevant for examining trial registration status (prospective or retrospective), our independent variable and selective outcome reporting in publication 'yes or no', our dichotomous variable. To perform a $\chi^2$ test, we conducted expected frequency summary statistics to see if the test was appropriate. A $\chi^2$ test was possible for our comparison between single-country and multicountry trials and for other comparisons, expected counts of most cells were less than five, so we proceeded to conduct a Fisher's exact test to test our assumptions. For time to publication, we used the, Kaplan-Meier survival analysis to determine time from trial completion to publication for the overall trials that had publication and for single-country versus multi-country trials. When performing time to publication analysis, we concentrated on the occurrence of publishing as some of the trials were ongoing while some trials had not started where they will give a future end date, therefore we could not censor. We also performed a Log-rank test to determine difference in time to publication between single-country and multicountry trials.

## RESULTS

We found 440 trials from the WHO-ICTRP (20 January 2019) and the Russian trials registry (30 March 2019) (figure 1). After eligibility assessment, 187 trials were excluded because they were duplicates of trials registered in more than one primary registry, or the purpose of a trial was not TB treatment, or they were a different study design such as observational study. In total, 253 trials were included in the analysis.

### Baseline characteristics
#### Recruitment centres
When exploring recruitment centres, we found that 30% (77) of the trials were multicountry trials, 22% (55) took place in China, 21% (53) in India, 13% (34) in South Africa, 9% (23) in Russia and 4% (11) in Brazil. Recruitment centres were identified as 'single-country single-centre' for 46% (117), 'multi-country multi-centre' for 30% (77) and 'single-country multicentre' for 24% (59) of the trials, respectively.

#### Registration status
Sixty-four per cent (163) of the trials were registered prospectively, 28% (69) retrospectively and 8% (21) did not report status at registration of the trial. When exploring registration status at country level, we found that prospective registration was 86% for multicountry, 79% for South Africa, 69% for China, 51% for Indian, 36% for Brazil and 4% for Russia. When we compared multicountry versus single-country trials, we found that multicountry trials more often registered prospectively compared with individual country trials (86% vs 55%, p=0.00) (table 1).

#### Sample size
The sample sizes of trials ranged from 0 to 1500 000 with a median sample size of 200 (IQR: 92–500), where one trial as identified from trial registry reported 0 as a sample size. At country level, trial sample sizes ranged from 20 to 6400 per trial with a median of 145 (IQR: 50–546) for Brazil; from 40 to 400 participants per trial with a median of 100 (IQR: 70–170) for Russia; from 0 to 4000 with a median of 175 (IQR: 110–327) for India; from 16 to 4176 with a median of 300 (IQR: 100–500) for China; and from 20 to 1500 000 with a median of 394 (IQR: 155–1155) for multi-country trials.

#### Age of participants
The majority of the trials (73% (183)) recruited adult participants, 6% (15) trials recruited adolescents and adults, 5% (13) trials included all ages, 4% (10) trials included children and adolescents and less than 1% (1) included children or adolescents only. Eleven per cent (30) of the trials did not report ages of participants. Out of the 183 trials that included adult participants only 30% (54) of the trials were those conducted in multiple countries, 24% from China, 22% (41) from India, 18% (32) from South Africa, 6% (10) from Brazil and 1% (2) from Russia. In the 30 trials that did not report ages of participants, 67% (20) were from Russia. Two trials that recruited children only and adolescents only, respectively, were multi-country trials.

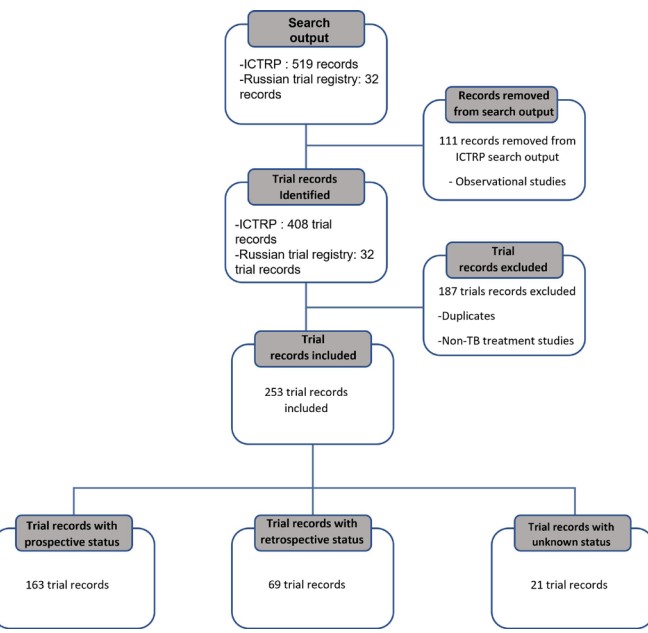

**Figure 1** Preferred Reporting Items for Systematic Reviews and Meta-Analyses flow diagram. ICTRP, International Clinical Trial Registry Platform; TB, tuberculosis.

**Table 1** Registration status of multicountry trials versus single-country trials

| Registration status | Multiple country trials | | Single-country trials | | Total | | P value |
|---|---|---|---|---|---|---|---|
| | % | n | % | n | % | n | |
| Not reported | 0 | 0 | 12 | 21 | 8 | 21 | 0.00 |
| Prospective | 86 | 66 | 55 | 97 | 64 | 163 | |
| Retrospective | 14 | 11 | 33 | 58 | 27 | 69 | |
| Total | 100 | 77 | 100 | 176 | 100 | 253 | |

## Type of treatment
### Categories of treatment
Trials evaluated different treatment, including 81% (207) studying drug treatment, 9% (22) studying behavioural treatment, 5% (12) nutritional supplementation, 3% (7) evaluating Direct Observed Therapy, almost 2% (4) evaluating surgical treatment and less than 1% (1) assessing rehabilitation for TB management. For the 207 trials evaluating the impact of drug treatment, 35% (73) were multicountry trials, 20% (41) conducted in China, 18% (37) conducted in India, 13% (28) from South Africa, 11% (22) from Russia and 3% (6) trials from Brazil. Drug treatment was further subdivided into the type of drug treatment comprising: (1) new combination treatment; (2) new drugs; (3) new treatment durations; and (4) new dosages as shown in table 2. When we compared single-country and multicountry trials, we found that drug treatment trials accounted for 76% of single-country trials and for 95% of multicountry trials (p=0.015).

We further analysed the drug treatment trials according to the phase of the trial and found that most were phase II (47), followed by phase III (41) and phase IV (20). However, most of the trials did not report their trial phase (68). The remaining 31 trials were similarly distributed in other phases of trials such as phase I (11), phase I/II (9) and phase II/III (11).

### Type of TB studied
Most of the included trials evaluated drug-sensitive TB treatment, 79% (201), followed by trials evaluating drug-resistant TB treatment, 21% (52). For the 201 trials studying drug-sensitive TB, 29% (58) were conducted in multiple countries, 23% (47) in India, 22% (44) in China, 12% (25) in South Africa, 8% (16) in Russia and 5% (11) in Brazil. In the 52 trials studying drug-resistant TB, 46% (24) trials conducted in multiple countries, 21%

(11) in China, 13% (7) in Russia, 12% (6) in South Africa and 8% (4) in India.

### Comorbidities and TB management
We found that 19% (47) of included trials evaluated TB treatment in participants with comorbidities. These included HIV/AIDS (42 trials), diabetes mellitus (3 trials) and other multimorbidities (2 trials). The two trials that studied TB treatment in participants with multimorbidities were conducted in India, while the three trials that studied TB treatment in participants with diabetes were conducted in China (2) and India (1). Fifty per cent (21) of trials that evaluated TB treatment in participants with HIV/AIDS were multicountries trials, 17% (7) were from India, another 17% (7) from South Africa, 5% (2) from China and the other 5% (2) from Brazil. Rheumatoid arthritis was studied as a comorbidity in one trial from China.

### Funding sources
When exploring the funding sources of included trials, we found that 25% (64) trials were funded by local organisations, 23% (58) were funded by international organisations, 17% (44) by the pharmaceutical industry, 9% (22) by universities, 6% (15) of the trials claimed to be self-funded and 5% (14) were funded by hospitals. We also found that 8% (19) had multiple funding sources while 7% (17) did not report funding sources (figure 2).

Funding at the country level in table 3 shows that majority of the multicountry trials were funded by international organisations (36) and industry (16), while for each of the trials funding was mostly by local and international organisation except for Russia, where 17 out of 23 were funded by industry. More single-country trials

**Table 2** Number of trials per type of drug treatment

| Type of TB drug treatment | n | % |
|---|---|---|
| New combination | 70 | 34 |
| New drug | 59 | 28 |
| New treatment duration | 45 | 22 |
| New dosage | 33 | 16 |
| Total | 207 | 100 |

TB, tuberculosis.

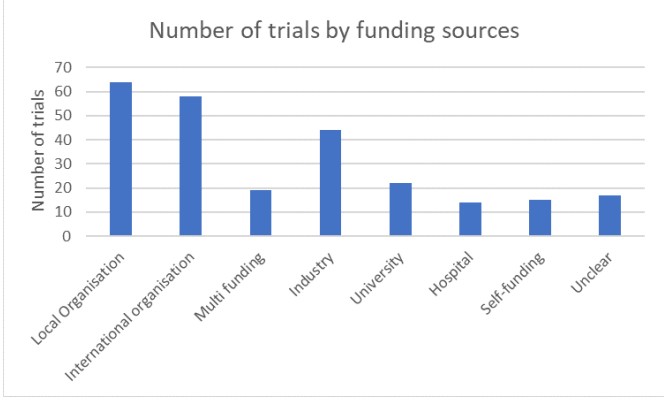

**Figure 2** Number of trials by funding sources.

**Table 3** Funding sources at country level

| Funding sources | Brazil n (%) | Russia n (%) | India n (%) | China n (%) | South Africa n (%) | Multicountries n (%) | Total number n (%) |
|---|---|---|---|---|---|---|---|
| International organisation | 1 (9) | 2 (9) | 7 (13) | 2 (4) | 10 (25) | 36 (47) | 58 (23) |
| Local organisation | 5 (46) | 3 (13) | 27 (50) | 15 (27) | 5 (15) | 9 (12) | 64 (25) |
| Self-funding | 0 (0) | 0 (0) | 4 (7) | 8 (14) | 2 (6) | 1 (1) | 15 (6) |
| Hospital | 1 (9) | 1 (4) | 0 (0) | 11 (20) | 0 (0) | 1 (1) | 14 (5) |
| Industry | 0 (0) | 17 (74) | 5 (9) | 1 (2) | 5 (15) | 16 (21) | 44 (17) |
| University | 3 (27) | 0 (0) | 5 (9) | 5 (9) | 5 (15) | 4 (5) | 22 (9) |
| Multiple funding | 0 (0) | 0 (0) | 4 (7) | 2 (4) | 6 (18) | 7 (9) | 19 (8) |
| Unclear | 1 (9) | 0 (0) | 1 (2) | 11 (20) | 1 (3) | 3 (4) | 17 (7) |
| Total number | 11 (100) | 23 (100) | 53 (100) | 55 (100) | 34 (100) | 77 (100) | 253 (100) |

compared with multicountry trials had local funding (31% vs 12%,) and multicountry trials had more international funding than single-country trials (47% vs 13%), p=0.00 (online supplemental appendix 2).

### Trial registration linked to publication
When focusing of publications linked to trial registration, we found that 156 trials that were completed 12 months or more by date were eligible to have a publication. Of these, we found that 65% (101/156 trials) were published and indexed in PubMed. At country level we found that of the 101 published trials, 43% (43) of the publications were from multicountry trials, 23% (23) from India, 16% (16) from South Africa, 13% (13) from China, 4% (4) from Brazil and 1% (2) from Russia.

### Selective outcome reporting
There was selective outcome reporting in 31 of 101 published trials as they reported a different primary outcome in the publication to that prespecified in the trial registry record (table 4). Seventy trials of 101 reported a primary outcome in the publication that matched the outcome prespecified in the trial registry record. We compared selective outcome reporting in prospectively registered trials versus in retrospectively registered trials and found that results were similar (30% vs 29%, p=0.318). When exploring the data across the BRICS countries we found that the multicountry trials had 19% of selective reporting versus single-country trials with 40% of selective reporting (p=0.029).

### Time to publication
Overall, the median time to publication from the anticipated completion of a trial according to the listed end date in the trial registry record was 26 months (range: 0–67 months). Multicountry trials publish within a median of 23 months (IQR: 12–34), which is similar to single-country trials 26 months (IQR: 15–37) (p=0.3117, Llog-rank test).

### DISCUSSION
Among the 253 included trials, we found that most trials (77) registered in primary registries were those with recruitment centres in multiple countries, followed by in China (55) and India (53). Of the BRICS countries, China and India were listed as countries with the highest burden of drug-sensitive TB and two of three countries with the highest burden in MDR-TB.[3] Results from this study show that these two countries account for the majority of the trials studying drug-sensitive TB and China with the second most trials studying drug-resistant TB following multicountries trials. Our results also found that many of the BRICS TB trials registered in primary registries are registered prospectively with a statistically significant difference between trial locations where multicountry trials are more likely to be registered prospectively as compared with single-country trials. This supports similar studies that have shown an increase in the number of prospectively registered trials.[23] Prospective registration of trials is recommended by the ICJME

**Table 4** Selective reporting for primary outcomes per country

| Primary outcome same as ICTRP | Brazil n (%) | Russia n (%) | India n (%) | China n (%) | South Africa n (%) | Multicountries n (%) | Total number (%) |
|---|---|---|---|---|---|---|---|
| No | 2 (50) | 2 (100) | 8 (35) | 6 (46) | 5 (31) | 8 (20) | 31 (30) |
| Yes | 2 (50) | 0 (0) | 15 (65) | 7 (54) | 11 (69) | 35 (81) | 70 (70) |
| Total | 4 (100) | 2 (100) | 23 (100) | 13 (100) | 16 (100) | 43 (100) | 101 (100) |

ICTRP, International Clinical Trial Registry Platform.

guidelines to reduce selective outcome reporting and publication bias in an event that a trial that has negative results.[20]

Even with an increase in prospective registration, we note that researchers, once they register their studies before trial begins, that records are rarely updated. We found a trial that had a sample of 0 which was a planned trial and not yet recruiting, however at the time of our analysis, that trial and many others were not updated despite their dates of enrolment and completion listed. If trial records are not updated with amendments, it leaves room for end users to question the transparency of the research being conducted and make assumptions about the trials.[24]

When exploring the ages of participants, we found that the majority (73%) of the trials studied adult participants only. While there are a few trials that studied all ages, only one trial studied children from birth to 12 years and one trial studying adolescents 13–17 years of age. This reiterates the substantial gap in TB treatment research done in children and adolescents where risk factor-related research has shown that children are the population most susceptible to transmission of TB and MDR-TB.[18] Children have remained poorly represented in clinical research due to, at times, challenging ethical and regulatory process of involving minors in clinical trials. This was recognised in 2009 prior to the launch Pan African Clincal Trial Registry (PACTR; https://pactr.samrc.ac.za/) child strategy which advocated for clinical research to be done in children to increase knowledge base of treatment where children are concerned.[25]

Eighty one per cent of the trials in the BRICS countries studied dug treatment trials, this likely indicates the efforts of these countries to find ways to treat and eradicate TB disease. According to research, there is evidence of effective TB treatment that has been widely adopted,[2] however trials in these countries are still focused on treatment trials for new drug approaches, but not enough emphasis on implementation such as behavioural or social aspects of TB including adherence. This is especially important to ensure that where there is availability of effective treatment for TB, the efforts are maximised to reduce TB epidemic. It would be of interest to explore in future research whether the research, trials and testing infrastructure that has been developed for COVID-19 pandemic, especially diagnostics such as PCR might have transferrable value to the TB domain and could offer capacity to support TB and MDR-TB research, trials and treatment in the future.

Our results showed that the majority of single-country trials were funded by local organisations whereas multi-country trials were likely to be funded by international funding, these results were statistically significant. Results also showed that majority of the trials received local ethics approvals with majority of the Principal Investigators (PIs) situated within the BRICS countries. This evidence supports research statement that suggest TB studies in the BRICS region tend to be funded and ran within the countries of

recruitment.[21] However, what is concerning is that 38% of the trials did not report having received ethics approval for conducting a trial, questions may arise about the credibility of these trials.

When exploring selective outcome reporting in publications, we found that 31 of 101 publications had selective outcome reporting. Results comparing selective outcome reporting by trial location showed that trials conducted in single countries were more likely to have selective outcome reporting than multicountry trials, these results were statistically significant. The results indicate that trials conducted in single countries which are most likely single-centre trials are prone to selective outcome reporting. Although the majority of published trials in our small sample have results aligned with their planned outcomes, 30% do not, which constitutes a substantial number and high level of selective outcome reporting. There was no statistically significant difference in selective reporting when comparing prospective and retrospective registration.

In a single-centre trial, sample sizes can be small and not enough to dilute differences that may influence outcomes as compared with multicentre trials which may lead to changes in outcome preferences.[25] This may point to the value in investing in larger trials that can be conducted in multicentre to improve outcomes reporting and generalisability of findings.

Lastly, for trials with publications, we explored the time from the presumed end date found in the registry to publication. The WHO and ICJME recommends publication within 12 months of trial completion.[19 20] We found that the majority (65%) of the trials completed 12 or more months by date had publication. Time to publication is on average 26 months and there is no significant difference between various factors such as the country of the trial, or whether a single-country trial which on average published at 26 months compared with multi-country trials which on average publishes at 23 months. Delays in study completion may be one of the reasons for delayed publication, but as registry records are rarely updated with actual study completion dates, this would require further exploratory research to understand the reasons for delays in depth.

The limitations of this study include that not all trials are registered or submitted to the WHO-ICTRP, some trials may have been missed. It is evident that researchers do not always update their trial information including anticipated end date and that may impact correct calculation of the Kaplan-Meier analysis; recruitment status may mislead the number of trials that are in fact completed but have not been updated, and there may be deviations from primary outcomes listed that are not updated in the trial registry record. Twenty more publications from included clinical trials since our initial search. Studies published after our analysis are unlikely to change the description analysis we have performed.

## CONCLUSION

TB treatment trial activity in BRICS countries, documented by our study, showed that the majority of registered trials were prospectively registered multicountry trials followed by country level nationally funded trials in China, India, South Africa and Russia. However, trials in children and adolescents were almost missing from these high-burden countries. The majority of trials are investigating drug-treatment, despite the recognised gap in research on TB treatment implementation. Time to publication from trial end-date was longer than 2 years, this lag time does not meet current WHO and ICMJE requirements to ensure research results reach the public and impact health policy timely. One third of the published trials had evidence of selective outcome reporting which raises concern and emphasises the need for researchers to update their trial information regularly once a trial registered to increase transparency, integrity and credibility of their research. Our mapping of TB treatment trials revealed the research gaps, which may guide policymakers, researchers and funders regarding the areas for focus to enhance TB treatment research. This is particularly true in the current situation in BRICS countries, where despite having effective treatment to manage TB to date, TB cases remain on the rise affecting many people.

**Author affiliations**
[1]Cochrane South Africa, South African Medical Research Council, Tygerberg, South Africa
[2]Department of Global Health, Stellenbosch University, Tygerberg, South Africa
[3]Biostatistics Unit, South African Medical Research Council, Tygerberg, South Africa
[4]Cochrane Russia, Russian Medical Academy for Continuing Professional Education of the Ministry of Health, Moscow, Russian Federation
[5]Kazan Medical University and RUDN University, Moscow, Russian Federation
[6]Statistics and Population Studies Department, University of the Western Cape, Bellville, South Africa
[7]Faculty of Medicine and Health Sciences, Division of Clinical Pharmacology, Department of Medicine, Stellenbosch University, Cape Town, South Africa

**Acknowledgements** The authors acknowledgethe South African Medical Research Council for unrestricted support of their work. This work was submitted to the Stellenbosch Unversity as a Masters thesis and was supported by Cochrane South Africa staff at South African Medical Research Council, Tamara Kredo, Elizabeth Pienaar, Duduzile Ndwandwe and Ameer Hohlfled. We also wish to thank Mikateko Mazinu and Esme Jordan for data analysis support from the South African Medical Research Council Biostats Unit.

**Contributors** LindiM led the development of the study, wrote the first draft of the protocol and paper, coordinated and integrated comments from coauthors. All authors will be guarantor of the paper. LovemoreM was involved in screening, data extraction and data analysis. DN and LEZ were involved in the screening, data extraction and translation of data from Russian to English. MM and EJ were involved in data analysis and provided support in the write up of the Data analysis and results section. DEN and TK critically provided supervision and mentorship to LindiM, together with DN and LEZ critically revised successive drafts of the manuscript, provided important intellectual input. All authors approved the final manuscript prior to submission.

**Funding** Grant awarded: Not Applicable Other funding: This work was supported by the South African Medical Research Council (through salaries for LindiM, DEN and TK).

**Competing interests** At the time of this work, LindiM, DEN, EP and TK were all part of the administrative team coordinating the Pan African Clinical Trials Registry (www.pactr.mrc.ac.za) and part of development of a new database for the South African National Clinical Trials registry. All authors have declared no competing interest.

**Patient and public involvement** Patients and/or the public were not involved in the design, or conduct, or reporting or dissemination plans of this research.

**Patient consent for publication** Not applicable.

**Ethics approval** This study was exempted from ethical consideration since the data collected is publicly available and at no point was confidential information from human participants used. When exploring whether ethical approval was obtained and where it was obtained from, we found that 50% (125) of the trials had obtained ethics approval locally, 4% (11) of the trials obtained approval only from international ethics review boards, 8% (20) of the trials obtained ethics approval from both local and international ethics review boards, while 38% (97) of the trials did not report ethics approval details.

**Provenance and peer review** Not commissioned; externally peer reviewed.

**Data availability statement** No data are available. Data are available upon reasonable request.

**ORCID iDs**
Lindi Mathebula http://orcid.org/0000-0002-4213-7318
Liliya Eugenevna Ziganshina http://orcid.org/0000-0003-1999-0705
Duduzile Edith Ndwandwe http://orcid.org/0000-0001-7129-3865
Tamara Kredo http://orcid.org/0000-0001-7115-9535

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
