## [Reviewer comments · BMJ Open]

ARTICLE DETAILS

TITLE (PROVISIONAL)	Planned, ongoing and completed tuberculosis treatment trials in Brazil, Russia, India, China, and South Africa: A 2019 cross-sectional descriptive analysis
AUTHORS	Mathebula, Lindi; Mapahla, Lovemore; Nurkhametova, Dilyara; Ziganshina, Liliya Eugenevna; Mazinu, Mikateko; Jordan, Esme; Ndwandwe, Duduzile; Kredo, Tamara

VERSION 1 – REVIEW

REVIEWER	Christine Tedijanto Harvard T.H. Chan School of Public Health
REVIEW RETURNED	24-Nov-2021

GENERAL COMMENTS	Thank you for this interesting paper summarizing planned, ongoing and completed tuberculosis trials in the BRICS countries. I think that this kind of descriptive analysis is crucial for understanding the current state of the literature, especially for a globally important topic such as TB treatment. In general, I thought the manuscript was well-written. I have several recommendations that I hope will help to clarify and improve the work: • As a general point, the term “selective reporting” seems rather strong to me as it seems to imply that the authors were purposely trying to mislead readers. Please feel free to maintain this term if it is widely accepted; otherwise, it may be worthwhile to adopt something more neutral like “mismatch between registered and reported primary outcome.”• The conclusions in the Abstract state that “only small fraction of which gets to publication.” I am assuming this conclusion is based on the 101 published trials out of 253 that were included in the analysis. However, after reading the methods, I am wondering if this is a biased reflection of the true publication proportion as some trials in the denominator may not yet be completed. An improved metric may be the number of published papers divided by the number of trials that are finished or should be finished.• If possible, it would be great to also analyze the 20 additional publications mentioned on page 3 as this would substantially increase the number of published studies (101 to 121?)• On page 7, it is noted that the most recent trial registration is used. Do the results differ if earlier versions of the registration are used? It may also be interesting to flag if the primary outcome is modified while the trial is ongoing.• On page 8, the last paragraph states that the analysis has four assumptions but only 2 are listed. Please clarify. Additionally, please provide rationale as to why we might expect selective outcome reporting for trials with multiple recruitment centres compared to single recruitment centres.
---

	 • On page 10, please clarify how a study could have a sample size of 0. • On page 12, please clarify the following sentence: “The remaining 31 trials were similarly distributed in other phases of trials.” • A PRISMA flow diagram may be helpful to describe selection of trials into the analysis even though this is not strictly a systematic review. • I am curious what the most common registered primary outcomes were, what the most common reported primary outcomes were, and whether there were outcomes that were more likely to be changed between registration and reporting. For example, were very rare outcomes most likely to be changed? It may also be worthwhile to see if the change between registered and reported outcome and potential reasons are noted in the paper discussion sections. • Is there any way to determine if the long time to publication could be partially attributed to delays in study completion (e.g. comparison of actual completion date to anticipated completion according to the trial registry record)? • Appendix 2 requires some cleaning / clarification. Please provide captions for each table so that it is clear what is being compared even if the column / row titles are shortened.
--	--

REVIEWER	Theolis Bessa Fundacao Oswaldo Cruz, Instituto Goncalo Moniz
REVIEW RETURNED	22-Dec-2021

GENERAL COMMENTS	Investigation needed to observe complications in anti-TB treatment compared to countries with higher disease burden
---

REVIEWER	Simon Brake NHS Walsall Clinical Commissioning Group
REVIEW RETURNED	14-Mar-2022

GENERAL COMMENTS	This was an excellent and useful paper. The only comment that I would offer for the discussion section is whether the research, trials and testing infrastructure that has been developed for COVID-19, especially e/qPCR might have transferrable value to the TB domain, and could offer capacity to support TB and MDR TB research, trials and treatment.
--

VERSION 1 – AUTHOR RESPONSE

Reviewer: 1

Dr. Christine Tedijanto, Harvard T.H. Chan School of Public Health

Comments to the Author:

Thank you for this interesting paper summarizing planned, ongoing and completed tuberculosis trials in the BRICS countries. I think that this kind of descriptive analysis is crucial for understanding the current state of the literature, especially for a globally important topic such as TB treatment. In general, I thought the manuscript was well-written. I have several recommendations that I hope will help to clarify and improve the work:

- As a general point, the term “selective reporting” seems rather strong to me as it seems to imply that the authors were purposely trying to mislead readers. Please feel free to maintain this term if it is widely accepted; otherwise, it may be worthwhile to adopt something more neutral like “mismatch between registered and reported primary outcome.”

Response: Thank you for this comment, may we assure you that the term “selective reporting” is widely accepted and should not mislead the readers. However, to ensure that this is clear for readers, this first time it is used, we include your neutral phrase in line 124, pg 5. In addition, we provide here a

reference to the Cochrane methodology materials, Collins's dictionary, BMC paper which are also included in the final manuscript:

<https://methods.cochrane.org/bias/reporting-biases>

http://methods.cochrane.org/sites/methods.cochrane.org.statistics/files/uploads/SMG_training_course_cardiff/2010_SMG_training_cardiff_day2_session3_dwan_altman.pdf

<https://www.collinsdictionary.com/dictionary/english/selective-reporting>

<https://systematicreviewsjournal.biomedcentral.com/articles/10.1186/s13643-015-0070-y>

- The conclusions in the Abstract state that “only small fraction of which gets to publication.” I am assuming this conclusion is based on the 101 published trials out of 253 that were included in the analysis. However, after reading the methods, I am wondering if this is a biased reflection of the true publication proportion as some trials in the denominator may not yet be completed. An improved metric may be the number of published papers divided by the number of trials that are finished or should be finished.

Response: Thank you for that observation, indeed we agree that analyzing for publication based on the entire sample may have been a biased reflection since other trials were not yet completed by date at the time of a cross-sectional analysis. We therefore re-analysed the data based on the trials that were supposed to be published 12 months from completion by date listed, this has been included in line 46-48 pg 2, line 369 to 371 pg 14, line 460 to 461 pg 18 and re-calculations in Appendix 2

- If possible, it would be great to also analyze the 20 additional publications mentioned on page 3 as this would substantially increase the number of published studies (101 to 121?)

Response: Thank you for this comment, our study is a cross sectional descriptive study, we analysed the data as available at the time of the study conduct. We think that the additional studies may not show much difference and as a team have considered whether an update evaluating similar TB data but during the COVID-19 pandemic may reflect any changes that have occurred since this cross-sectional analysis was undertaken.

- On page 7, it is noted that the most recent trial registration is used. Do the results differ if earlier versions of the registration are used? It may also be interesting to flag if the primary outcome is modified while the trial is ongoing

Response: Thank you for that observation. We chose the latest record by date as we believed any changes, missing information and amendments to their earliest trial submission and registration would be corrected in their final trial registration. We agree that this concept would be interesting to explore, however may be outside of the scope of this analysis and not likely to change the findings.

- On page 8, the last paragraph states that the analysis has four assumptions but only 2 are listed. Please clarify. Additionally, please provide rationale as to why we might expect selective outcome reporting for trials with multiple recruitment centres compared to single recruitment centres.

Response: Thank you for that observation, we initially had four assumptions and had corrected to test for two assumptions, see line 243, pg 9.

Rationale: in cases of multiple recruitment centres chances for flaws with data collection, management and reporting might be higher, but this is only an assumption, hence the need to test it.

- On page 10, please clarify how a study could have a sample size of 0.

Response: Thank you for that observation. We used data directly sourced from WHO's ICTRP. One trial record as found in the ICTRP data has listed zero as a sample size and that trial has not been updated. See line 292 pg 11. We have also included in discussion the observation of trials not being updated line 408 to 414 pg 16.

- On page 12, please clarify the following sentence: “The remaining 31 trials were similarly distributed in other phases of trials.”

Response: Thank you for the comment, this was meant to indicate that the rest of the trials were distributed to the other phases, phase 1, phase 1/2, phase 2/3 in similar numbers for each these have now been included in line 328, pg 12

- A PRISMA flow diagram may be helpful to describe selection of trials into the analysis even though this is not strictly a systematic review.

Response: Thank you for the comment. A Prisma flow has been included as figure 1

- I am curious what the most common registered primary outcomes were, what the most common reported primary outcomes were, and whether there were outcomes that were more likely to be changed between registration and reporting. For example, were very rare outcomes most likely to be changed? It may also be worthwhile to see if the change between registered and reported outcome and potential reasons are noted in the paper discussion sections.

Response: Thank you very much for this comment and suggestion. This is indeed interesting to look into the problem of selective outcome reporting in further depth. However, this is beyond the scope (objective) of our current research and submitted manuscript, we analysed the trial and outcome numbers across BRICS countries and discrepancies between trial registration and reporting information. The suggested approach would be a new objective for a new, different research project.

- Is there any way to determine if the long time to publication could be partially attributed to delays in study completion (e.g. comparison of actual completion date to anticipated completion according to the trial registry record)?

Response: Thank you very much for this comment. There is unfortunately no way to link long time publication to delays in study completion even if we do assume it is mostly the case, as we only rely on the information provided in the trial records by the researchers and if that information is not updated then end users on the information. See line 408 to 414 and line 464 to 467 pg 16 and 17.

- Appendix 2 requires some cleaning / clarification. Please provide captions for each table so that it is clear what is being compared even if the column / row titles are shortened.

Response: Thank you very much for this observation. Appendix 2 has been cleaned and captions have been provided.

Reviewer: 2

Dr. Theolis Bessa, Fundacao Oswaldo Cruz

Comments to the Author:

investigation needed to observe complications in anti-TB treatment compared to countries with higher disease burden

Response: Thank you very much for this comment, we agree that it would be indeed interesting to look into the problem of adverse effects of anti-TB treatment, and particularly in comparative analysis of BRICS countries to the countries with higher disease burden. This idea may become a subject for a new research project.

Reviewer: 3

Prof. Simon Brake, NHS Walsall Clinical Commissioning Group, University of Warwick

Comments to the Author:

This was an excellent and useful paper. The only comment that I would offer for the discussion section is whether the research, trials and testing infrastructure that has been developed for COVID-19, especially e/qPCR might have transferrable value to the TB domain and could offer the capacity to support TB and MDR TB research, trials and treatment.

Response: Thank you very much for this comment. We agree that it would be indeed interesting to assess the contribution of COVID-19 newly established infrastructure to the TB research domain. This is beyond the scope of the current research, however, we will add a comment to reflect this in the discussion as follows “It would be of interest to explore in future research whether the research, trials and testing infrastructure that has been developed for COVID-19 pandemic, especially diagnostics such as PCR might have transferrable value to the TB domain and could offer the capacity to support TB and MDR TB research, trials and treatment in the future”(line 431 to 435, pg 17).